Evidence for low prevalence of ranaviruses in Ontario, Canada’s freshwater turtle population

Carstairs Sue J. suecarstairs@sympatico.ca
Ontario Turtle Conservation Centre (Kawartha Turtle Trauma Centre) , Selwyn , Ontario , Canada
Colla Sheila
Electronic publication date: 2019 May 20
Publication date: 2019
Volume: 7
Electronic Location ID: e6987
Received 2019 Feb 25; Accepted 2019 Apr 19
Copyright: ©2019 Carstairs
Copyright year: 2019
Copyright holder: Carstairs
License: This is an open access article distributed under the terms of the Creative Commons Attribution License, which permits unrestricted use, distribution, reproduction and adaptation in any medium and for any purpose provided that it is properly attributed. For attribution, the original author(s), title, publication source (PeerJ) and either DOI or URL of the article must be cited.
License URL: https://creativecommons.org/licenses/by/4.0/

Keywords: Ranavirus, Ranavirus turtles, Ranavirus prevalence, Ranavirus ontario canada, Ontario turtles, Ontario turtles ranavirus prevalence

Funding: The Elsa Foundation The Margolis Foundation The J.P. Bickell Foundation This work was supported by The Elsa Foundation, The Margolis Foundation, and The J.P. Bickell Foundation. The funders had no role in study design, data collection and analysis, decision to publish, or preparation of the manuscript.

==============================
Background

Ontario, Canada is home to eight native species of turtles; all eight are federally listed as Species At Risk, due to anthropogenic threats. However, until recently, reports of infectious disease have been lacking. Ranavirus is seen as an emerging threat for ectotherms globally, with mass die-offs most often reported in amphibians. Ranavirus has been detected in Ontario’s amphibian populations, can be transmitted via water, and can be transmitted from amphibians to turtles. However, no studies on the prevalence of this virus in Ontario’s turtles have previously been carried out. With recent reports of two confirmed positive case of ranavirus in turtles in Ontario, a knowledge of the ecology of ranavirus in Ontario’s turtles has become even more important. This study estimates the prevalence of ranavirus in Ontario’s turtles, and investigates the hypothesis that this is a newly emergent disease.

Methods

Sixty-three samples were tested for ranavirus via PCR. These included a variety of turtle species, across their home range in Southern Ontario. Fifty-two of the samples originated from the liver and kidney of turtles who had succumbed to traumatic injuries after being admitted to the Ontario Turtle Conservation Centre; ten of the samples were taken from cloacal swabs, lesion swabs, or tail clips collected from live turtles showing signs of clinical disease. One of the live turtles was later euthanized for humane reasons and PCR was also carried out on the liver/kidney.

Results

None of the 63 samples were found to be positive for ranavirus via PCR. The zero prevalence found in this study translates into a population prevalence estimate of less than 5%, with no change in prevalence from 2014–2018.

Discussion

This is the first report on the prevalence of ranavirus in Ontario’s turtles, and will help build an understanding of the ecology of this virus in Ontario. Ranavirus has historically been underreported in reptiles, but there has been an increase in global reports recently, most likely due to increased awareness. A carrier state is thought to exist in reptiles which makes surveillance in the population via random sampling a viable method of detection of prevalence. The first report of ranavirus in Ontario turtles occurred in 2018. This study suggests a continued low population prevalence for the years 2014–2018, however. Ongoing surveillance is necessary, as well as investigation of the eDNA presence in waterways as compared to the PCR of resident turtles, to further understand the sensitivity of these species to ranavirus infection. The utilization of qPCR would be helpful, to better quantify any positives encountered.

Introduction

The genus Ranavirus belongs to the family Irodiviridae; DNA viruses that infect invertebrates and ectothermic vertebrates such as insects, fish, amphibians and reptiles, crustaceans and molluscs (Williams, Barbosa-Solomieu & Chinchar, 2005). Ranaviruses have been shown to be capable of infecting fish, amphibians and reptiles (Mao, Hedrick & Chinchar, 1997) and have since been found in at least 175 species of fish, amphibians and reptiles, across 52 families, on all continents except Antarctica (Gray & Chinchar, 2015). Frog Virus 3 (FV3, genus Ranavirus) is one of a number of ranavirus species recognized and has been called the “type species” of the genus Ranavirus (Mao, Hedrick & Chinchar, 1997). Most ranavirus infections found in reptiles so far have been FV3- like, and in the USA, only FV3-like viruses have been detected in reptiles (Huang et al., 2009; Allender et al., 2011). However, it has been shown that other ranavirus species can infect reptiles (e.g., Huang et al., 2009; Stohr et al., 2015). Infection with ranavirus can result in high morbidity and mortality in amphibians, reptiles and fish, and has been recognized as being responsible for widespread die-offs of ectothermic vertebrates since the 1990s (Gray, Miller & Hoverman, 2009). In Ontario, Canada, ranavirus has been responsible for significant mortality events in amphibians (Greer, Berrill & Wilson, 2005; Duffus & Andrews, 2013). It can be passed between amphibians, reptiles and fish (Bandin & Dopazo, 2011) and can be transmitted through the water between amphibians and turtles (Gray et al., 2014; Brenes et al., 2014). As a result, amphibians have been suggested as potential reservoir hosts for susceptible chelonians (Johnson et al., 2008). It is also possible, however, that turtles are acting as a reservoir host for other species (Brenes et al., 2014) with their apparent higher tolerance for infection aiding this.

The prevalence of the virus, host susceptibility, and severity of disease has not been previously studied for turtles in Canada. In addition, it is unknown whether species other than FV3 may infect these turtles. Ranavirus was reported for the first time in reptiles in Ontario, Canada, in 2019 (McKenzie et al., 2019). This was also the first time this has been identified in a common snapping turtle (Chelydra serpentina). A second positive case was also found in 2018, in a wood turtle (Glyptemys insculpta) (Canadian Wildlife Health Cooperative blog 2018). In the USA, the majority of reported cases of ranavirus have involved the eastern box turtle (Terrapene carolina carolina) (e.g., De Voe et al., 2004; Allender et al., 2011; Winzeler et al., 2018); a study on its prevalence in Eastern painted turtles was carried out in Virginia, USA (Goodman, Miller & Ararso, 2013).

Ontario Turtle Conservation Centre (OTCC; operating name of the Kawartha Turtle Trauma Centre) annually admits approximately 1,000 injured or ill native Ontario turtles from across their home range in Southern Ontario, for treatment, rehabilitation and subsequent release. Ontario’s eight species of turtles are all considered At Risk federally and include two globally endangered species (the spotted turtle, Clemmys guttata and Blanding’s turtle, Emydoidea blandingii). Ontario’s turtle populations face many threats, including habitat loss and fragmentation, illegal harvesting for the pet trade, and road mortality (Gibbons et al., 2000), but reports of disease in wild turtles have been lacking. Emerging diseases remain a potential concern for these species, due to the heavy pressures already placed on their populations. It has been suggested that wildlife rehabilitation centres can be important biomonitors of ecosystem health (Sleeman, 2008). OTCC serves as a useful early monitoring point for prevalence of individual diseases, as well as disease outbreaks. With the report of the first case of ranavirus in turtles in Ontario, determining whether this is a novel emerging disease, or a preexisting disease previously unsurveyed, is essential since the two have very different potential consequences for the population. This study commences to test the hypothesis that this is a recent emerging disease, by evaluating ranavirus prevalence over four years, to evaluate whether an increase is being seen. This study reports results for ranavirus testing (FV3) by polymerase chain reaction (PCR) in a variety of turtle species across the province of Ontario. Random screening of pooled liver and kidney samples of turtles that succumbed to their injuries after admission to our hospital was carried out between 2014 and 2018. In addition, PCR testing for FV3 was carried out via cloacal swab, lesion swab, or tail clipping, on any admitted turtle that exhibited suggestive clinical signs or any illness of unknown cause.

Materials & Methods

Turtles are admitted to the OTCC hospital (Kawartha Turtle Trauma Centre) from across their home range in Ontario, and beyond. The majority of admissions are due to road injuries, but OTCC also admits those found with any clinical signs of disease. While approximately 1,000 turtles are admitted and treated annually, under Ministry of Natural Resources and Forestry Wildlife Custodian Authorization number 20025217, not all survive. The turtles who succumb to their injuries provide an opportunity to collect organ tissue for subclinical disease testing. Sixty-three turtles admitted to OTCC from Ontario, Canada (Fig. 1) were tested for ranaviral DNA using PCR. Fifty two of the tests were carried out on kidney and liver samples of a random sample of turtles that had succumbed to traumatic injuries between 2014 and 2018; ten tests were carried out on live turtles showing symptoms of disease between 2017 and 2018. One of these was later euthanized for humane reasons (by Sue Carstairs; College of Veterinarians of Ontario license 3649), and the liver and kidney also tested.

Figure 1 Location of turtles tested for ranavirus by PCR in Ontario, Canada.

Red, Painted turtle, Chrysemys picta; Green, snapping turtle, Chelydra serpentina; Purple, Blanding’s turtle, Emydoidea blandingii; Yellow, map turtle, Graptemys geographica. Wood turtles, Glyptemys insulpta have been omitted for confidentiality reasons. Map data@OpenStreetMap contributors, ODbL.

Table 1 Species, sex, clinical signs and case outcomes for live turtles with suggestive clinical signs admitted to the OTCC from across Ontario, Canada and tested for ranavirus via PCR.

Species	Clinical signs	Outcome	
Snapping turtle	Necrotic stomatitis; found on snow in January	Ranavirus PCR/Herpesvirus consensus PCR-negative by tail clipping, lesion swab and cloacal swab	
Male		Aeromonas sp. cultured	
		Full recovery with supportive treatment and antibiotic therapy	
Wood Turtle	Part of a tracking study; found underweight, and with nasal exudate plugging nares	Ranavirus PCR/Herpesvirus consensus PCR negative by cloacal swab	
Juvenile 550 grams		Nares clear on presentation.	
		Increased weight and improved condition with supportive care	
Snapping turtle	Ulcerative dermatitis of head and neck	Ranavirus PCR/Herpesvirus consensus PCR negative by cloacal swab	
Male 6.8 kg	Marked dehydration		
	Oral/nasal exudate with increased upper airway sounds	Full recovery and subsequent release, with supportive care and antibiotic therapy	
Painted turtle	Bilateral keratitis	Ranavirus PCR/Herpesvirus consensus PCR negative by cloacal swab.	
Female 483 grams	Anorexia	Died in care	
Wood turtle	Left eye nonvisual	Ranavirus PCR/Herpesvirus consensus negative by cloacal swab	
Juvenile 415 grams		Deemed likely traumatic injury; released	
Snapping turtle	Bilateral blepharitis	Ranavirus PCR negative by cloacal swab	
Male 10.5 kg	1 × 1 cm mass near lateral canthus of left eye	Mass surgically removed and histopathology identified fibrous tissue	
		Released	
Snapping turtle	Neurological signs: torticollis to the right and circling to the right	Ranavirus PCR negative by cloacal swab	
Juvenile 195 grams		Poorly responsive; assumed to be traumatic head trauma	
		No improvement; euthanasia carried out for humane reasons by veterinarian Dr. Sue Carstairs licence number 3649	
Painted turtle	Blepheredema left eye, edematous neck region, dsypnea	Ranavirus PCR/Herpesvirus consensus PCR negative by cloacal swab	
Female 563 grams	Poor mentation	Died in care despite supportive care and antibiotic treatment	
	Blood smear showed a regenerative response in the Red Blood Cell line, as well as marked toxic changes to 100% of the heterophils, increased density lung field left side	Subsequent PCR for ranavirus on liver/kidney, found negative	
		Lung abscess identified on post mortem	
Wood turtle		Ranavirus PCR negative by cloacal swab	
	Respiratory signs seen by biologists studying	PCR Herpes virus positive; Gleptemys herpesvirus by DNA sequencing	
Snapping turtle	Generalized edema, lethargy, anemia, anorexia	Ranavirus PCR negative by cloacal swab	
Male 8.5 kg			

Many of these were also tested for herpes virus via PCR (VanDevanter et al., 1996). Table 1 shows these turtles and their clinical signs. Samples from live turtles were collected from cloacal swabs, swabs of suggestive lesions and in one instance, tail tip in addition to swabs. Swabs provided by StarswabTM Multitrans Collection and Transport system (Starplex Scientific Inc., 50 Steinway Boulevard, Etobicoke Ontario Canada M9W 6Y3. Cat. No. S160) were utilized, and immediately placed in the viral transport medium provided. Samples from deceased turtles were collected from liver and kidney post mortem using aseptic technique. Samples were stored in sterile containers containing no additives, and immediately frozen. Frozen samples were sent to Idexx laboratories (1345 Denison Street, Markham, Ontario, Canada L3R 5V2) for transport to the Animal Health Centre of British Columbia (1767 Angus Campbell Road, Abottsford, British Columbia, V3G 2M3) for PCR testing. Samples were evaluated for ranavirus infection; Frog virus 3 (FV3), using polymerase chain reaction (PCR), with the primers targeting the major capsid protein (MCP) gene with an amplicon size of 482 bp (base pair). The primers were FV3-MCP-1′-F (5′-GCA GGC CGC CCC AGT CCA-3′) and FV3-MCP-2-289 R (5′-GGG CGG TGG TGT ACC CAG AGT TGT-3′). The amplicon can be sequenced from a positive PCR to determine the virus strain. The target segment was amplified in 25 uL of commercial mix (IllustraTM puReTaq Ready- 96 To-GoTM Beads, GE Healthcare UK, Limited) with 800 nM each of forward primer and reverse primer, nuclease-free water and 2 ul of DNA template. Samples were run on thermal cycler (Tetrad2, BioRad Laboratories, Montreal, QC, Canada) using a thermal cycling program as follows; initial denaturation for five minutes at 95C, followed by 40 cycles of 95C for one minute, 61C for one minute, 72C for one minute and a final extension of 72C for seven minutes. DNA was extracted using QiaAMP DNA Mini kit (Qiagen Inc, Toronto, Ontario) according to the manufacturer’s instructions. DNA was handled using standard molecular biology protocols. After extraction, DNA was stored at 4C. In most cases, DNA extractions and PCR setups are carried out on the same day. DNA is stored at −30C long term. ISO/IEC 17025 and AAVLD (American Association of Veterinary Laboratory Diagnosticians) Standards for quality assurance and quality control, are followed. Conventional PCR products were run on a 2% agarose gel with ethidium bromide and analyzed using a uV photo documentation system (Alphalmager HP Imaging System, ProteinSimple, Sata Clara, Ca, USA). Positive FV3 PCR results would be confirmed by direct sequencing of the 482 base pairs (bp) amplicon. Analytical sensitivity of this PCR is approximately 3,000 copies of the genomic DNA. The positive control used was an iridovirus isolate received from the University of Saskatchewan. DNA sequencing of the PCR amplicon identified the positive control as epizootic haematopoietic necrosis virus.

Reptile ranaviruses have a multispecies host range, and therefore reptile-specific PCR assays are not required (Wirth et al., 2018).

The population prevalence estimate was obtained using an approximation of the formula presented in Cameron & Baldock (1998) and summarized by Cannon & Roe (1982).

Results

None of the 63 samples were found positive for ranavirus FV3, by PCR in the liver and kidney homogenate from deceased turtles, or cloacal swabs/lesion swabs/ tail clipping of live, ill turtles. One sample was found positive for Herpesvirus (identified as Gleptemys herpesvirus by DNA sequencing). The sample results came from a random sampling of turtles across their home range in Ontario, Canada, as well as those exhibiting suggestive clinical signs. In addition, they included multiple species, and were taken from 2014–2018. While we are aware of ranavirus presence in the turtle population of Ontario (McKenzie et al., 2019), the prevalence appears from this study to be low. Based on a sample size of 63, we can be 95% confident that the population prevalence is less than 5%. Given an extremely sensitive test, and no known bias for sampling turtles with versus without disease, this estimate appears reasonable. The lack of positive cases in these results suggest a continued low prevalence and possible preexistence of the disease prior to its’ first discovery in 2018.

Discussion

It is believed that ranavirus has historically been underreported in reptiles (Daszak et al., 1999; Johnson et al., 2008; Allender, 2012) but reports are increasing, probably due to increased awareness. Increased surveillance, improved testing methods, and emergence of ranavirus infection, could also be responsible. The first two reported cases of ranavirus infection in wild turtles in Ontario, Canada, likely also represents the result of increased awareness, since testing has not been previously carried out in this province. It is possible that the virus has been present in turtles in Ontario for some time; the OTCC has seen cases showing characteristic lesions for a large number of years but did not commence testing until relatively recently. The clinical signs of ranavirus infection in reptiles can be variable. Ranavirus has been responsible for high mortality in turtles, with one group of Mediterranean tortoises (Testudo Graeca) reported to have 100% mortality (Marschang et al., 1999). Clinical signs include necrotizing stomatitis, esophagitis, fibrinous and necrotizing splenitis, and multicentric fribrinoid vasculitis (Johnson et al., 2008), and can include necrotizing tracheitis and pneumonia. Obvious and nonspecific external lesions include marked blepharitis, and cervical edema (Miller et al., 2015). Clinical signs are not pathognomonic for ranavirus and include sudden onset of severe illness or sudden death with no premonitory signs (Allender, 2011). Signs can appear similar to those of other infectious agents such as mycoplasma and herpesvirus infections, bacterial infection secondary to trauma, as well as non-infectious issues such as Vitamin A deficiency. Evidence also suggests that reptiles can be asymptomatic carriers of ranviruses (e.g., Goodman, Miller & Ararso, 2013; Goodman, Hargadon & Carter, 2018).

Prevalence of ranavirus in the USA has been found to be variable. PCR on oral-cloacal swabs and tail clips were used to survey two species in three water bodies in Virginia; the Eastern painted turtle, Chrysemys picta picta, and the Common musk turtle, Sternotherus odoratus (Goodman, Miller & Ararso, 2013). They found a prevalence of 4.8–31.6% in painted turtles, and zero in musk turtles. Studies on 140 free-ranging Eastern box turtles (Terrapene carolina carolina) admitted to rehabilitation centres in the USA, and 39 free-ranging turtles, showed 0–3% prevalence of ranavirus by PCR on blood and oral swabs (Allender et al., 2011). Johnson et al. (2008) screened for the prevalence of iridovirus in free-ranging gopher tortoises (Goperus polyphemus) and Eastern box turtles in the USA, via indirect enzyme-linked immunosorbent assay (ELISA) on plasma samples. Overall prevalence in gopher tortoises was 1.5%, and 1.8% in Eastern box turtles. However, the duration of antibody response is unknown in these species, and turtles may fail to sero-convert, or may die prior to being surveyed, resulting in an underestimated prevalence. Due to its potentially devastating effects on already declining turtle populations, it is important to further our knowledge of ranavirus ecology in Ontario’s turtle populations, as a potential novel emerging disease. Questions to answer include the prevalence of infection, susceptibility to disease and severity of disease, as well as the presence or absence of a subclinical carrier state. The first reported case of ranavirus in Ontario (McKenzie et al., 2019) showed classical clinical signs of ranavirus; marked bilateral palpebral swelling, conjunctival ulceration, ulcerative stomatitis. The turtles possessing clinical signs that were tested by the Ontario Turtle Conservation Centre, also showed similar signs; including necrotizing stomatitis, cervical edema, palpebral swelling, and ulcerative dermatitis. PCR on oral-cloacal swabs and tail clippings, have been shown to yield false negatives for ranavirus when compared to organ testing, such as liver (Gray, Miller & Hoverman, 2012; Goodman, Miller & Ararso, 2013). Gray, Miller & Hoverman (2012) feel that this non-lethal test is still useful for surveillance, however. In addition, sensitivity has been shown to increase as time post exposure increases (De Voe et al., 2004), with tail-clip samples converging on whole-animal homogenates if infection has been present for at least five days. The obvious chronicity of the illnesses of the live turtles in our study, suggested a duration of many weeks (as indicated by body condition and hematological parameters). As a result, had the lesions been caused by ranavirus, our testing methods should have had a high sensitivity to detect the virus via PCR on the live turtles. Johnson, Pessier & Jacobson (2007) found similar detection levels for ranavirus via PCR, from oral vs cloacal swabs, in red eared sliders (Trachemys scripta elegans). Cloacal swabs were used in our study, for logistical reasons, except where oral lesions were present. The PCR testing of liver and kidney on the deceased turtles, holds a high sensitivity for detection of prevalence of subclinical infection, and is the preferred test site.

This is the first study of the ecology of ranavirus in turtles of Ontario. The prevalence in this study was zero, with a population estimate of less than 5% prevalence if we assume that the test is perfectly sensitive. Prevalence rates in chelonians have historically reflected mortality rates (Allender, 2011), which supports this estimate. We cannot be sure there is not an unknown bias in capturing infected vs uninfected turtles, however, or that positive cases have not died before capture. Since prevalence rates appear constant for the past 5 years, it is possible that this disease has not newly emerged but has been present in turtles in Ontario for as long as its presence in amphibians, however with more resilience to mortality. Environmental DNA-based quantification (eDNA) of ranavirus infection has been suggested to be a useful and non-invasive method of ranavirus detection in wildlife and aquaculture (Hall et al., 2016). Knowledge of the prevalence and titre of the virus in Ontario’s water bodies would allow us to start to discover Ontario turtle’s susceptibility to infection. Overlaying our results over eDNA results in associated water bodies, would greatly augment knowledge as to susceptibility of infection in resident turtles. If eDNA is found in significant amounts and yet the prevalence in turtles is still low, it suggests that these turtles may have a low susceptibility to infection. In addition, use of qPCR would allow quantification of any positive results acquired.

Conclusions

Our study is the first to study the prevalence of ranavirus in Ontario’s turtles. The OTCC has the unique opportunity to gather samples from turtles across their home range in Ontario and should act as an accurate bioindicator for this disease. Random screening for subclinical disease, as well as screening of those showing suggestive clinical signs, act in concert to provide useful monitoring for ranavirus prevalence. The current results are encouraging as to indicating a continued low population prevalence, with no change in prevalence seen over the years 2014 to 2018. It was feared that the cases seen in 2018 might indicate a marked increase in number of cases to follow. With no further cases seen in 2018, it suggests that this disease may have been present in the population for some undetermined time; possibly as long as its presence in amphibians but had not been identified due to lack of testing. The consequences to turtle populations are more favorable for this scenario than for the alternative of a newly emergent disease with potential catastrophic population effects. However, if is shown that turtles can harbor the disease in a subclinical state, this might indicate a significant source of infection for species more vulnerable to the virus. Commencing to understand the ecology of ranavirus in Ontario’s turtles is vital in planning conservation strategies for these and other ectotherms vulnerable to this disease. An ongoing screening process is essential, both of random samples from across the province, and from any turtles that die of unknown causes, or that are showing suggestive clinical signs. We hope to shed further light on the potential for recovery from disease if infected, and the presence or absence of a subclinical carrier state in these species, as well as their potential to act as a reservoir for other ectotherms.

Supplemental Information

File S1 Detailed description of all turtles tested for ranavirus

Click here for additional data file.

Thanks go to Tomy Joseph, veterinary virologist at the Animal Health Centre, BC Ministry of Agriculture, for the processing of the samples and provision of methods of same. I would like to thank Margaret Chan and Amanda Klack for the collection of the majority of the organ samples utilized in this study.

Additional Information and Declarations

Competing Interests

Author Contributions

Animal Ethics

Data Availability

Sue Jacqueline Carstairs is employed as the Executive and Medical Director of the Ontario Turtle Conservation Centre (Kawartha Turtle Trauma Centre).

Sue J Carstairs conceived and designed the experiments, performed the experiments, analyzed the data, contributed reagents/materials/analysis tools, prepared figures and/or tables, authored or reviewed drafts of the paper, approved the final draft.

The following information was supplied relating to ethical approvals (i.e., approving body and any reference numbers):

This research was conducted under the Ministry of Natural Resources and Forestry Wildlife Custodian Authorization number 20025217, which authorizes operation of a wildlife rehabilitation facility, and all procedures this entails. Veterinary procedures were carried out by Dr. Sue Carstairs, licenced under the College of Veterinarians of Ontario (licence number 3649).

The following information was supplied regarding data availability:

The raw data is available as a Supplementary File. This includes identification of all individuals tested, including species, sex, and location found.

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
