# Peer review of "Evidence for low prevalence of ranaviruses in Ontario, Canada’s freshwater turtle population"

_PeerJ, doi:10.7717/peerj.6987_

## Round 0.1 · original submission · Minor Revisions

Please address the comments from the 3 reviews. I feel they will greatly ameliorate your manuscript prior to publication. I look forward to receiving the edited manuscript.

·

Basic reporting

Overall the author used clear and concise English throughout the text. The references provided throughout the text are sufficient and provide the necessary background information to the reader. The article is structured appropriately for the target journal. The figures and tables referenced in the paper are included in the submission, however Figure 1 and Table 1 referenced in the Materials and Methods section are incorrectly labeled at the end of the text. Additional comments on wording, cited literature, figures and tables can be found in the comments to author portion of this review.

Experimental design

The article has a clear, well defined objective of determining the prevalence of ranavirus in free-ranging turtles in Ontario, Canada. The research is novel in that no such survey has been conducted previously meaning little is known regarding the impact ranavirus has on chelonians in Ontario, Canada. The methods used are described well enough that the study could be replicated.

Validity of the findings

The results are clearly presented. I commend the author for writing this article, despite not detecting ranavirus in any of the samples tested. Surveys which result in no detection are likely under-reported in amphibian disease literature.

Additional comments

Line 40-42: The conclusions sentences found on these lines are very similar to what is repeated in line 43 of the Methods portion of the abstract. Re-wording the last two sentences of the Background section of the abstract might remove some redundant text.
Line 74: Please provide a citation for the statement regarding FV3 occurrence in reptiles.
Line 82: Brenes et al. 2015 should be Brenes et al. 2014
Line 82: I understand why you mention amphibians as reservoirs here, but you might want to add a statement suggesting turtles are also a potential reservoir for ranaviruses since they can tolerate infection. The Brenes et al. 2014 paper cited in the previous sentence suggest turtles and fish as reservoir species.
Line 83-84: Suggest changing “susceptibility for infection” to “host susceptibility”
Line 87: common snapping turtle
Line 89: eastern box turtle (Terrapene carolina carolina)
Line 91: rephrase ; ranavirus prevalence was determined to be 4.8%-31.6% and 0% for eastern painted turtles (Chrysemys picta) and common musk turtles (Sternotherus odoratus) respectively in Virginia, US.
Line 95: consider spelling all numbers <10 throughout the text (e.g. 8 to eight)
Line 110: Turtles are admitted to the OTCC hospital
Line 116: Consider changing: Sixty-three turtles submitted to OTCC from Ontario, Canada (Figure 1) were tested for ranaviral DNA using PCR;
Line 120: Consider rephrasing: During necropsy or evaluation each turtle was observed for clinical signs of disease (Table 1).
Line 123: Consider adding a Cat# for the swabs
Line 129: Consider changing “viral testing” to “PCR testing”
Line 130-134: The only section of the methods which needs more description is found in these lines. Please provide what kit was used to extract the DNA from your samples. Also, please describe whether positive and negative controls were run on each gel.
Line 140: Rephrase: “liver and kidney pooled samples” to “liver and kidney homogenate from deceased turtles”
Line 151: “Ontario Turtle Conservation Centre” should be OTCC after the first mention.
Line 158: A citation needed at the end of this sentence. You could possibly cite the “Comparative Pathology of Ranaviruses and Diagnostic Techniques” section of the “Ranaviruses” book published in 2015. DOI: https://doi.org/10.1007/978-3-319-13755-1
Line 159: Consider changing “mimic” to “appear similar to”. Using mimic would be an anthropomorphism
Line 164-166: Either ignore my previous comment for Line 91 or remove this section.
Line 168: (Terrapene Carolina Carolina) to (Terrapene carolina carolina)
Line 171: (Goperus Polyphemus) to (Gopherus polyphemus)
Line 174: “may die before surveyed” to “may die prior to being surveyed” ; “,so prevalence may be underestimated.” to “resulting in an underestimated prevalence.”
Line 208: “Our facility” to “The OTCC”
Table 1: is actually Figure 1; make sure all scientific names are italicized and spelled appropriately.
Table 2: is actually Table 1; caption needs more description. (e.g. “Species, sex, case number, clinical signs and case outcome for turtles submitted to the OTCC which were tested for ranavirus via PCR.”

Reviewer 2 ·

Basic reporting

I find the paper to be well written, however I have some comments on the reporting of the work.

I think the title of the article does not do a good job of encapsulating the results that follow. I would consider changing the title to something like "Evidence for low prevalence of ranaviruses in Ontario's freshwater turtle populations."

I believe that given the likely readership and the narrow scope of this paper, the amount of detail afforded to the general biology and pathology of ranaviruses during the introduction could be deemed excessive. I would consider tightening up the introduction to focus primarily of ranavirus prevalence and the important of understanding pathogen prevalence in wildlife diseases.

My personal opinion is that this work would be better suited for presentation as a shorter format communication in a reputable journal specialising in wildlife diseases. Given that this is essentially a survey, the work does not seem to be based on any particular testable hypothesis. If the authors do wish to publish in PeerJ then I would like to see the introduction reworked to include some sort of hypothesis declaration and clearer rationale for the study.

Experimental design

The experimental design is straightforward and appears to be sufficient for the purposes of this study.

However, there is a lot of missing information regarding the molecular diagnostics employed for this study.

Specifically:

How was DNA extracted from each sample type?

How was DNA subsequently handled?

Why did the authors decide to use conventional PCR rather than one of the available quantitative PCR assays?

How was the positivity, or not, of a PCR test deduced? I assume this was by electrophoresis, however it would be useful to state this for clarification.

Were positive controls included in each PCR / Gel?

Were samples extracted / amplified in duplicate or triplicate, as is standard in diagnosis the presence of a pathogen using PCR of qPCR.

At present the molecular portion of the work is not replicable and as such this section requires significant attention before this work can be published in its current form.

Validity of the findings

The current finding of the paper is zero prevalence of ranavirus in the turtles sampled.

I believe that this paper would be strengthened by incorporating the methods alluded to in lines 195 - 197 of the discussion into the paper methods so that paper can include the finding of a suggested prevalence less than 5% in turtles in Ontario. This is a potentially more impactful statement to make.

Whilst there is nothing wrong with presenting a negative detection story, in order for it to be interesting of a generalist journal such as PeerJ, the authors should take care to present the results in a positive framework.

Additional comments

I find the paper to be well written. However, I feel that the methodologies are seriously lacking in detail and that the presentation of the papers results lacks ambition.

It is my personal opinion that this work would find a more suitable home in a wildlife diseases journal rather than a generalist journal like PeerJ.

Reviewer 3 ·

Basic reporting

Minor grammatical errors within sections - please see the annotated PDF for suggested changes for strength and clarity. Otherwise, the article meets the standards for literature references, background, article structure, and relevant results for a general survey (not hypothesis-driven, but important for prevalence and disease studies".

Experimental design

Minor additions would be helpful within the methods section, specifically concerning PCR protocol. The research fits within the Aims and Scope of the journal, with a meaningful research question.

Validity of the findings

Statistical results were first reported in the discussion section- they should be moved appropriately, but they are justified and helpful to the reader. Negative results are important for prevalence studies, and are well-supported in the discussion and conclusion section.

Additional comments

Overall, I think this paper is useful for the epidemiology of ranavirus in chelonians. With minor edits as suggested in the attached PDF, it will be a good piece to have available, especially if ranavirus should become problematic in the region in the future and to enhance our evaluation of the disease dynamics at a population and geographic scale.

Annotated reviews are not available for download in order to protect the identity of reviewers who chose to remain anonymous.

---

## Round 0.2 · accepted · Accept

Many thanks for clearly addressing the reviewer comments.

#